# Effects of Silicon Content and Tempering Temperature on the Microstructural Evolution and Mechanical Properties of HT-9 Steels

**DOI:** 10.3390/ma13040972

**Published:** 2020-02-21

**Authors:** Junkai Liu, Wenbo Liu, Zhe Hao, Tiantian Shi, Long Kang, Zhexin Cui, Di Yun

**Affiliations:** 1Department of Nuclear Science and Technology, Xi’an Jiaotong University, Xi’an 710049, China; k611478@stu.xjtu.edu.cn (J.L.); liuwenbo@xjtu.edu.cn (W.L.); chierjj@stu.xjtu.edu.cn (Z.H.); sttian@stu.xjtu.edu.cn (T.S.); czx1013@stu.xjtu.edu.cn (Z.C.); 2Institute of Modern Physics, Chinese Academy of Sciences, Lanzhou 730000, China; kanglong@impcas.ac.cn

**Keywords:** ferritic/martensitic steel, tempering temperature, Si content, mechanical properties, microstructure

## Abstract

Two kinds of experimental ferritic/martensitic steels (HT-9) with different Si contents were designed for the fourth-generation advanced nuclear reactor cladding material. The effects of Si content and tempering temperature on microstructural evolution and mechanical properties of these HT-9 steel were studied. The microstructure of experimental steels after quenching and tempering were characterized by X-ray diffraction (XRD), scanning electron microscopy (SEM), and transmission electron microscopy (TEM); the mechanical properties were investigated by means of tensile test, Charpy impact test, and hardness test. The microscopic mechanism of how the microstructural evolution influences mechanical properties was also discussed. Both XRD and TEM results showed that no residual austenite was detected after heat treatment. The results of mechanical tests showed that the yield strength, tensile strength, and plasticity of the experimental steels with 0.42% (% in mass) Si are higher than that with 0.19% Si, whereas hardness and toughness did not change much; when tempered at 760 °C, the strength and hardness of the experimental steels decreased slightly compared with those tempered at 710 °C, whereas plasticity and toughness increased. Further analysis showed that after quenching at 1050 °C for 1 h and tempering at 760 °C for 1.5 h, the comprehensive mechanical properties of the 0.42% Si experimental steel are the best compared with other experimental steels.

## 1. Introduction

Advanced 9%–12%Cr ferritic/martensitic (F/M) steels including HT-9 (12Cr-1MoVW) have been considered as one of the most promising candidate reactor fuel cladding and structural materials in Generation IV advanced nuclear reactors, owing to their superior thermomechanical properties, irradiation resistance (void swelling and embrittlement) and corrosion resistance at elevated temperatures [1,2,3,4,5,6,7,8,9,10]. In the past few decades, a large amount of HT-9 steels have been widely studied all over the world. However, there are still some serious challenges for the usage of HT-9 in future reactors: neutron irradiation leads to an increase in the ductile–brittle transition temperature (DBTT) and will further affect its fracture resistance considerably; insufficient creep strength and fracture toughness at high temperature [11]. Consequently, the comprehensive mechanical properties of the current HT-9 steels need to be further improved. There are mainly three ways to strengthen 9%–12%Cr ferritic/martensitic steels: solution strengthening, precipitation strengthening and dislocations strengthening, and the essence of steel strengthening mechanism is to provide better resistance to the slip of dislocations. All these methods mainly depend on the chemical composition and the heat treatment of the steels. The heat treatment processes of HT-9 and other ferritic/martensitic steels have been studied by many researchers [12,13,14,15,16,17,18,19,20]. From the recent research on the heat treatment effects, the results show that the tempering temperature is the most important factor affecting the yield strength and elongation of HT-9 steel [21]. As a result, adjusting the tempering temperature is an effective method to optimize the mechanical properties of HT-9 steel.

Adjusting the element composition is another effective way to enhance the comprehensive properties of ferritic/martensitic steels. The effects of tungsten [22], tantalum [23], zirconium [24], etc. on mechanical properties of ferritic/martensitic steels were studied by different researchers. It was reported that the addition of silicon can improve the oxidation resistance [25,26], the liquid metal corrosion resistance [27,28] of 9%–12%Cr ferritic/martensitic steels. However, on the mechanical properties, silicon has two conflicting effects. Silicon can improve the strength and toughness due to the refinement of grain size and its solution-hardening effect [29,30]. At the same time, Si is a ferrite formation element, and excessive addition of silicon will result in the increase of chromium equivalent and the decrease of nickel equivalent correspondingly, which will decrease the single-phase zone of austenite and form δ-ferrite easily in the process of austenitizing and further affect the microstructure and mechanical properties of the steels. There is also controversy on how the δ-ferrite influences the mechanical properties of ferritic/martensitic steels. It was demonstrated by many researchers that massive δ-ferrite has a detrimental effect on the mechanical properties [31] and δ-ferrite is believed to cause significant increase in the ductile–brittle transition temperature (DBTT) in ferritic/martensitic steels [2,32,33,34]. On the contrary, some researchers drew the opposite conclusion that the soft δ-ferrite can increase the toughness and the ductility of steels [35]. Another work showed that when the δ-ferrite content is less than 1%, it will have positive effects [33]. So the effects of δ-ferrite phase on the impact properties and strength of 12%Cr ferritic/martensitic steel need to be further discussed and understood. Moreover, it’s important to find a balance between the corrosion resistance and the mechanical properties (e.g., yield strength, the hardness, and the toughness) based on adjusting the Si content for HT-9 steel. On the premise that the mechanical properties will not be influenced, Si element should be added to improve the corrosion resistance.

In the present work, we designed two kinds of HT-9 experimental steels to investigate how the Si content and the tempering temperature affect the mechanical properties of these experimental steels. The study is based on the characterization of microstructure of the steels with different Si content and performing tensile tests, Charpy impact tests and hardness tests at room temperature using standard samples. After a detailed explanation of the experimental procedures, the experimental and test results are outlined and discussed. The reasons for the differences of mechanical properties are also explained from the point of view of the different microstructures.

## 2. Experimental

The main elemental compositions of the steels examined in this study are listed in Table 1, and the contents of As, B, Ca, Co, P, S, and Zn are less than 0.005% in wt.%. It can be observed from Table 1 that, there are no significant differences about the element content between these two experimental steels except for the Si content. Different detailed heat treatment conditions for both steels are summarized in Table 2. With the different heat treatment conditions, four kinds of samples were obtained. The samples were fabricated into regular shapes with 10 mm × 5 mm × 5 mm and 8 mm × 5 mm × 5 mm for optical microscopy (OM) and scanning electron microscopy (SEM) characterization, whereas the tensile specimens were designed according to China national standard GB/T228-2010 and the Charpy V-notch impact specimens were designed according to China national standard GB/T229-2007. The schematic of the two types of specimens are shown in Figure 1.

The phase content of as-received specimens were analyzed by the X-ray diffraction (XRD) examination on a Bruker D8 ADVANCE (Bruker, Karlsruhe, Germany) for determining whether there was remaining austenite, and the 2θ angle range is 20° to 100°, the scanning step is 0.02°.

The microstructure of the steels were characterized by the Nikon MA200 inverted metallographic system optical microscope (Nikon, Tokyo, Japan) and Gemini 500 scanning electron microscopy (SEM) (Zeiss, Jena, Germany) after mechanical grinding, polishing and etching in Villela reagent (1 g picric acid + 5 mL hydrochloric acid + 100 mL ethanol). The transmission electron microscopy (TEM) was applied to better observe the internal microstructure of the experimental steels with a JEM 200-CX facility (JEOL, Tokyo, Japan). The samples for TEM were mechanically grinded and polished to a thickness of ~100 μm, and then 3 mm TEM discs were then punched out, followed by twin-jet-electropolishing in a solution of perchloric acid + ethanol (1:10) at 20 V and −20 °C.

The tensile test was carried out on the electronic universal testing machine (Instron 3382, Instron, Canton, OH, USA) with a stretching strain rate of 1 × 10^−3^ s^−1^ at room temperature. The Charpy impact test was conducted on pendulum impact testing machine (CEAST RESIL 5.5, CEAST, Torino, Italy) at room temperature to evaluate the toughness of the experimental steels (the impact energy is 400 J and impact velocity of the pendulum is 5.4 m/s). After tensile and impact tests, the fractographies of steels were characterized by the SU3500 SEM (HITACHI, Tokyo, Japan). The microhardness of the steels was measured by HV-1000 microhardness tester (Veiyee, Laizhou, China) with a load of 50 g and a load time of 10 s.

## 3. Results

### 3.1. Microstructures

Figure 2 shows the XRD spectrum of the four kinds of samples. It can be observed that, after quenching and tempering, there is only body-centered cubic structural ferritic/martensitic phase in every sample but no retained austenite.

The metallurgical structure pictures of each sample are shown in Figure 3. It exhibits that the prior austenite grain size of the tested steels are all about 90 μm measured by the linear intercept method, and there is no big difference between different samples. Only the silicon content and the tempering temperature between four samples are different. The silicon content of the ferritic/martensitic steel was reported that could affect the Ac_3_ (at which the transformation of ferrite to austenite is completed) and further affect the grain size [29,30]. From the research on the ferritic/martensitic steel by Anya [30], 0.31 wt.% silicon steel has the lowest transformation temperature. When the content of silicon in steel lower than 0.31 wt.% or higher than 0.31 wt.%, the Ac_3_ will increase. In this experiment, the silicon content 0.19 wt.% in 1# steel is lower 0.31 wt.%, whereas 0.42 wt.% in 2# steel is higher than 0.31 wt.%, so the difference of Si content of 1# and 2# steels will not significantly influence the grain size. Also, the tempering temperature will not affect the grain size. Therefore, after heat treatment, the grain size of prior austenite of four kinds of steels will not have big difference. After quenching and tempering, the 0.19 Si-steels exhibit full martensite phase shown in Figure 3a,b, and the martensitic lath is thin and long thanks to its superior hardenability; the 0.42 Si-steels in Figure 3c,d, by contrast, contain minimal bright irregular shaped δ-ferrite phase which is distributed at the prior austenite grain boundary. The δ-ferrite just appears at very few regions. The average δ-ferrite content of the 2L and 2H samples are approximately 0.425% and 0.203% respectively counted from three images at different locations of the samples by the Image Pro Plus software and the size of δ-ferrite is about 10 μm on average, which is significantly smaller than the prior austenite grain size.

High-magnification SEM was used to investigate the martensite and ferrite microstructure in more detail. Figure 4a–d displays the martensitic lath of each sample. The lath boundaries are seen to be decorated with M_23_C_6_ carbides and there distributed the coarse M_23_C_6_ and fine MX precipitates in the matrix, which is consistent with others’ findings in ferritic/martensitic steels [36]. With increasing tempering temperature, the density of randomly distributed fine precipitates in the martensitic matrix decreases and the size of carbides at the martensitic boundary increases. This indicates that the dispersed carbide particles aggregated and grew under higher tempering temperature. The δ-ferrite, which is observed only in 2# (0.42 Si) steel sample, is shown in Figure 5. It is observed that heavy-gauge carbides were distributed preferentially along martensite/δ-ferrite and δ-ferrite/δ-ferrite boundaries. There are also small white particles inside the δ-ferrite grain which are identified as dislocations rather than carbides according to the research work of Abe [2], and these small white particles will be further discussed together with the TEM characterization results.

The more detailed investigation of the TEM foils is shown in Figure 6. The size of the martensitic lath of the experimental steels is not uniform and the width of the lath is between 100 and 500 nm. The precipitates at grain boundaries and in the matrix are identified as M_23_C_6_ and MX phase, respectively, by selected area electron diffraction (SAED) in the 1L steel. There is almost no intragranular fine precipitate in the 1H and 2H steels as seen in Figure 6d,f. The number density and size distribution of M_23_C_6_ in the four experimental steels are counted with an average of five images from different locations and the results are shown in Figure 7. The size of lath boundary precipitates in the 1H and 2H (high temper temperature) steels are larger than those in the 1L and 2L (low temper temperature) steels, whereas the density of precipitates show the opposite trend. This result suggests that there is disappearance of the fine MX phase and coarsening of the M_23_C_6_ phase with increasing tempering temperature, and the result is in line with SEM characterization. As seen in Figure 6a,e, there are massive dislocation tangles and fine and small dislocations distributed in the martensitic matrix, whereas in Figure 6d,f, the martensitic matrix has fine substructures with subgrain boundaries and dislocation lines, the dislocation tangles and fine dislocations almost disappeared. The sub-grain boundaries and dislocation lines are formed by the interaction of dislocations at high tempering temperature. Figure 8a,c displays the δ-ferrite structure of the 2L and 2H steels, respectively. The δ-ferrite is irregularly shaped and surrounded by high density precipitates at the boundaries between δ-ferrite and martensite, and the carbides are identified as M_23_C_6_ by SAED in Figure 8b,d. Note that the average diameter of carbides at δ-ferrite boundaries is ~100 nm and is smaller than that at the martensitic lath boundaries according to Figure 9, which is different from the results of other researchers that the δ-ferrite can lead to the production of larger precipitates [32]. The reason why the carbides around the δ-ferrite are smaller and different from other researchers’ results is that the content of the δ-ferrite is too low and the grain size of δ-ferrite is too small to provide enough Cr and C elements for the growth of the carbides around the grain boundaries of δ-ferrite in this experiment. These fine carbides in the current experiments will be beneficial for the mechanical properties of ferritic/martensitic steel. No obvious precipitates were observed inside the δ-ferrite. Besides, dislocations saturated in the δ-ferrite confirming that the white particles in the SEM images are dislocations rather than precipitates.

### 3.2. Mechanical Tests Results

Figure 10 gives a summary of results obtained in the tensile tests, Charpy impact tests, and hardness tests on the experimental steels. With an increase in the tempering temperature, the yield strength (YS), ultimate tensile strength (UTS), and hardness decrease slightly, whereas the reduction of area, elongation, and impact energy increase. This indicates that the ductility and impact toughness are enhanced with high tempering temperature. Higher Si content leads to an increase in the YS, UTS, reduction of area, and elongation, whereas it results in little change in impact energy and hardness. This suggests that the higher Si content can improve the strength, ductility, stiffness of the experimental steels, but have no significant effects on the impact toughness and hardness.

### 3.3. Fractography

Figure 11 and Figure 12 show the typical micro-view of fracture surface of the four experimental steels after Charpy impact and tensile tests at room temperature, respectively. From the impact fracture morphology (Figure 11), all experimental steels are typical ductile fracture featured with a lot of dimples varying in size and depth, along with inclusions, and second-phase particles distributed at the bottom of dimple, suggesting that all the samples have excellent toughness at room temperature. There is no notable difference between the four fractographies which is consistent with the close impact absorbed energy of each sample. As seen in Figure 12, the tensile fracture morphology of the low temperature tempered samples (Figure 12a,c) shows a mixture of ductile fracture with ductile dimples and locally quasi-cleavage fracture with tear ridges (indicated by the black arrow). When the tempering temperature increases (Figure 12b,d), the samples show totally ductile dimples surface suggesting ductile fracture. The results of tensile fracture morphology are coincident with the higher elongation and percentage reduction of area with high tempering temperature. It indicates that the high temperature temper can improve the ductility of the experimental steels.

## 4. Discussion

### 4.1. Effects of Si Content on Microstructure and Mechanical Properties

In the work of Chen [29] and Anya [30], the Si content affects the mechanical properties of ferritic/martensitic steels mainly through the increase of Ac_3_ temperatures (at which the transformation of ferrite to austenite is completed), which lead to the decrease of prior austenite grain size. Grain refinement caused by higher Si content further optimizes the mechanical behaviors. Also, Si can promote the formation of Laves phases through decreasing the solubility of Mo [37] and W [29] in 9Cr ferritic/martensitic steels. In this work, there is no grain refining or formation of Laves phases, so the Si element affects the mechanical properties mainly through the solution-hardening effect and the formation of δ-ferrite in the steels. In the low-carbon steels, Si is a substitutional solute atom, the solution-hardening effect of Si improves the yield strength of the steels from 41.3 to 96.5 MPa per wt.%, whereas increases the impact transition temperature [38,39]. The solution-hardening effect of Si is beneficial to the yield strength and hardness but degrades the impact toughness in the experimental ferritic/martensitic steels. In the present study, yield strength was found to increase with Si content at a rate of 104.23 (low tempering temperature) and 137.33 (high temper temperature) MPa per wt.%, which is larger than that in low-carbon steels, and there is no big difference about the impact energy with the increase of Si content. Therefore, the formation and the content of δ-ferrite influenced by Si element is also an important factor for the mechanical properties of ferritic/martensitic steels.

In the HT9 steels, the C, Mn, and Ni elements are austenite formers, whereas the Si, Cr, W, V, and Nb are ferrite formers. Increasing Si content will increase the chromium equivalent and decrease the nickel equivalent. Thus, during the process of austenitizing, the single phase region of austenite decreases and δ-ferrite is easily formed. As for the influence of the δ-ferrite on the mechanical behaviors, most researchers agree that the presence of ferrite can lower the ultimate tensile strength, yield strength [35,40] and impact toughness [12,33,41]. There are several ways that δ-ferrite degrades the mechanical properties of ferritic/martensitic steels: (1) The cracks nucleate in the δ-ferrite preferentially because the δ-ferrite is softer than the martensitic matrix [33,41]. (2) The formation of brittle, coarse M_23_C_6_ on the ferrite grain boundary and the interface between the ferrite and the matrix because of the low solubility of carbon and the high solubility of chromium in δ-ferrite [33,41,42]; the microcracks are easier to nucleate at the places of larger secondary carbides along the δ-ferrite compared with those in the martensite matrix, so the toughness and strength of steels will be degraded [32]. (3) More grain boundary regions are generated by the formation of δ-ferrite, and thereby more extra space were produced for carbides to precipitate. Therefore, in the matrix, there are fewer carbides precipitated because total quantity of precipitated carbides remain unchanged [32].

However, in this experiment, the higher Si content improves the YS and the UTS of the experimental steels. On the one hand, the strengthening regularity of steel containing ferrite receives the combined effects of ferrite softening and fine grain strengthening [1]. The ferrite content of the high Si experimental steels in this experiment is very small (<1%) and can only be observed in some localized areas of the samples. Besides, the grain size of the ferrite is much smaller than that of the martensite. Eventually fine grain strengthening effect of δ-ferrite becomes strong enough to counteract its softening effect. At the same time, the solution-hardening effect of silicon in steels could also contribute to the improvement of strength. On the other hand, there are large amount of carbides formed at the boundary between ferrite and martensite matrix, and the dimensions of these kind of carbides have not coarsened and they are even smaller than the M_23_C_6_ at martensitic lath boundary according to Figure 9. This is mainly because the content and the size of δ-ferrite are small. Therefore, these fine carbides could pin the interface and significantly improve the interface stability and inhibit the uneven deformation of the interface. As a result, the carbides around δ-ferrite rather improve the plasticity, strength of ferritic/martensitic steels by the pinning effects [33,35,43] than produce a substantial embrittlement effect [34]. As for the hardness and toughness of the experimental steels, on the one hand, the hardness and toughness of δ-ferrite are lower than the martensitic matrix [33,41], and the presence of δ-ferrite causes further degradation to the properties of the steels; on the other hand, the toughness and hardness are enhanced by the fine carbides around the δ-ferrite. Also, considering the solution-hardening effect of Si, these factors happen to offset each other, so the impact toughness and hardness of experimental steels are not affected greatly by the Si content eventually. Because of the softening effect of δ-ferrite and the fine precipitates produced by δ-ferrite, the ductility of experimental steels is enhanced with the increase of Si content.

In this work, Si content of 0.42 wt.% fraction has just exceeded the critical value of Chromium equivalent to form minimal δ-ferrite and form fine carbides at the interface between δ-ferrite and martensitic matrix, eventually leading to the best mechanical properties of such steel among the experimental steels.

### 4.2. Effects of Tempering Temperature on Microstructure and Mechanical Properties

There are two main effects of the tempering temperature on the microstructure and mechanical properties of ferritic/martensitic steels. On the one hand, the tempering temperature could impact the precipitates on martensitic lath boundaries and inside the matrix. The fine precipitates in martensitic lath and prior austenitic boundaries can increase the strength of the experimental steels by preventing dislocation slip. In this work, with increasing tempering temperature, the fine carbides aggregate and grow and the solute atoms around the carbides precipitate. The coarsening and growth of precipitates reduce the effect of solution strengthening and dispersion strengthening. As a result, with the increase of tempering temperature, the strength and hardness of the experimental steels decrease. The elongation, section shrinkage, and impact energy (plasticity and toughness) increase slightly because the dislocations are easier to glide without the pinning effect of the fine carbides; on the other hand, the distribution and behavior of dislocations in martensitic matrix are affected by the tempering temperature. At low tempering temperature, there are mainly dislocation tangles and fine dislocations in the martensitic matrix, and the dislocation tangles brought by the interaction of dislocations could effectively prevent the movement of dislocations. Consequently, the strength and hardness of the steels are improved. With increasing tempering temperature, dislocation tangles in the martensitic matrix are replaced by the dislocation lines and sub-grain boundaries which are easier to glide. Therefore, the hardness and strength are impaired while the plasticity and toughness are ameliorated.

The HT-9 experimental steels were designed as reactor cladding and structural materials, which requires not only outstanding mechanical strength, but also plasticity and toughness. After quenching at 1050 °C for 1 h and tempering at 760 °C for 1.5 h, the YS and the TS of high Si steels are relatively high and the plasticity and toughness are excellent, the fractographies of tensile and impact tests show all dimples fracture surface (Figure 11b,d and Figure 12b,d) and therefore the comprehensive mechanical properties of the 2H steel are the best. As a result, the 2H steel was chosen to further conduct corrosion and irradiation tests in the near future.

## 5. Conclusions

The effects of silicon content and tempering temperature on the mechanical properties of HT-9 ferritic/martensitic steels have been studied by Charpy impact test, tensile test, hardness test, and microstructural analysis. The following conclusions can be drawn.

(1) The content of Si affects the mechanical properties by solution-hardening effect and formation of δ-ferrite. First, the solution-hardening effect of Si enhances the strength and hardness but degrades the impact toughness in the experimental ferritic/martensitic steels. Second, the δ-ferrite is softer than the martensitic matrix which degrades the strength, hardness and the toughness but it is beneficial to the ductility. Meanwhile, the small ferrite grain size could reinforce the steels. Third, the fine carbides precipitated at the interface between the δ-ferrite and martensite optimize the mechanical properties. Eventually, with the combination of all these effects, the strength and the ductility are improved by the increase of Silicon content while the toughness and hardness show no big change.

(2) The tempering temperature can affect the mechanical properties of the experimental steels by influencing the precipitates and dislocations. With the increase of tempering temperature, the fine precipitates coarsen and depress the pinning effect and the solution strengthening effect. At the same time, the fine dislocations and dislocation tangles disappeared and the dislocation line and sub-grain boundaries formed. As a result, the strength and hardness decrease while the ductility and toughness increase.

(3) After heat treatment, all the experimental steels showed excellent strength, ductility, toughness and hardness. The 0.42 wt.% Si steels after 1050 °C quenching and 760 °C tempering possess the best comprehensive mechanical properties considering the application to the reactor fuel cladding and structural materials.

## Figures and Tables

**Figure 1 materials-13-00972-f001:**
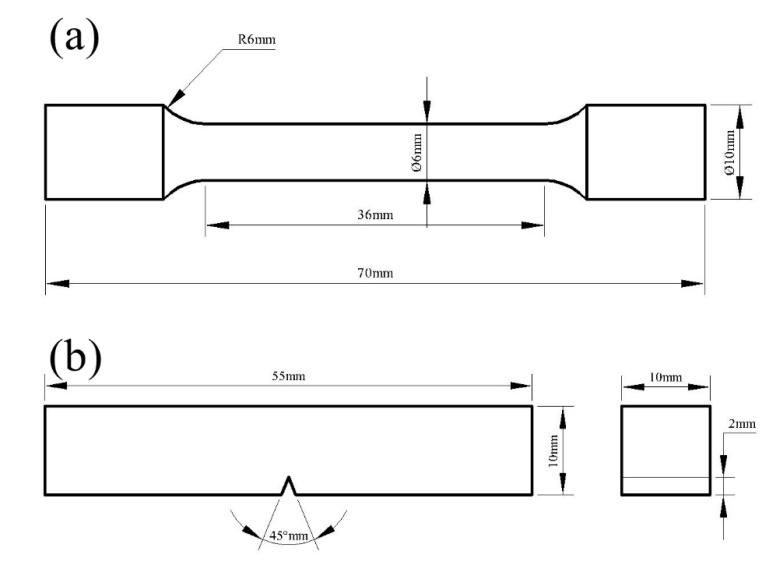
Shape and dimensions of tensile specimen (**a**) and Charpy shock specimen (**b**).

**Figure 2 materials-13-00972-f002:**
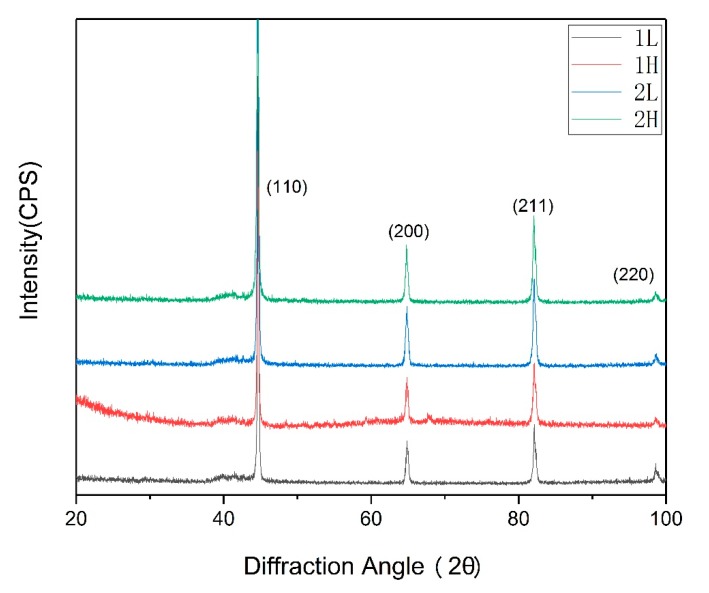
XRD diagram for experimental steels.

**Figure 3 materials-13-00972-f003:**
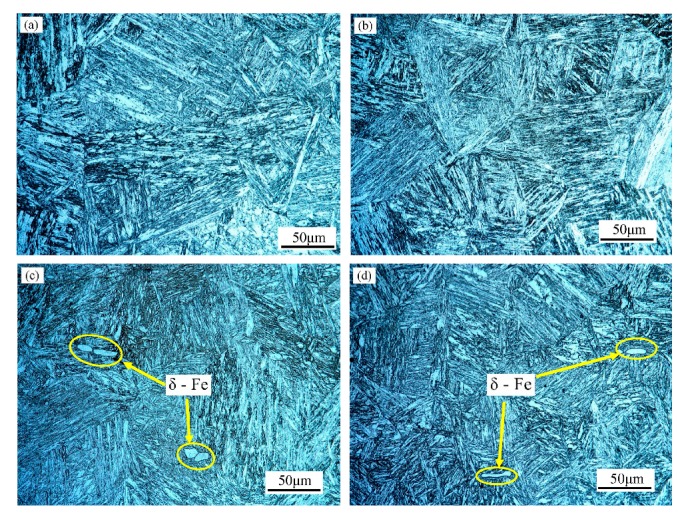
Optical microstructures of the experimental steels after heat treatment (**a**) 1L (low Si and low temperature), (**b**) 1H (low Si and high temperature), (**c**) 2L (high Si and low temperature), and (**d**) 2H (high Si and high temperature).

**Figure 4 materials-13-00972-f004:**
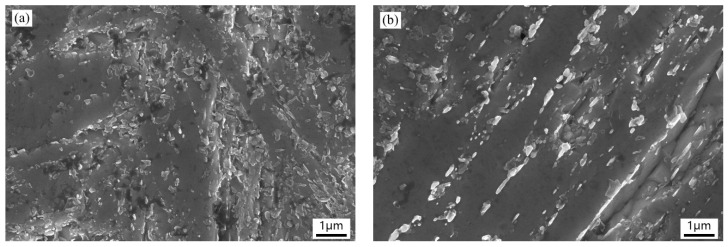
SEM images of the experimental steels after heat treatment (**a**) 1L (low Si and low temperature), (**b**) 1H (low Si and high temperature), (**c**) 2L (high Si and low temperature), and (**d**) 2H (high Si and high temperature).

**Figure 5 materials-13-00972-f005:**
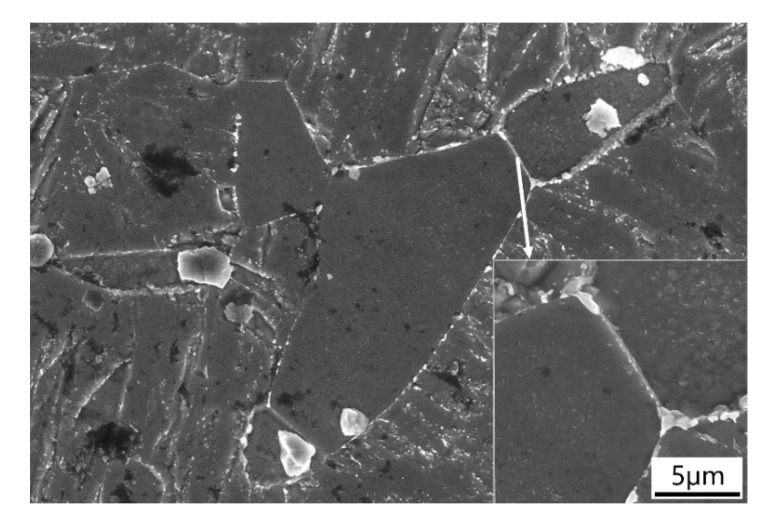
SEM image of δ-ferrite in 2# steel.

**Figure 6 materials-13-00972-f006:**
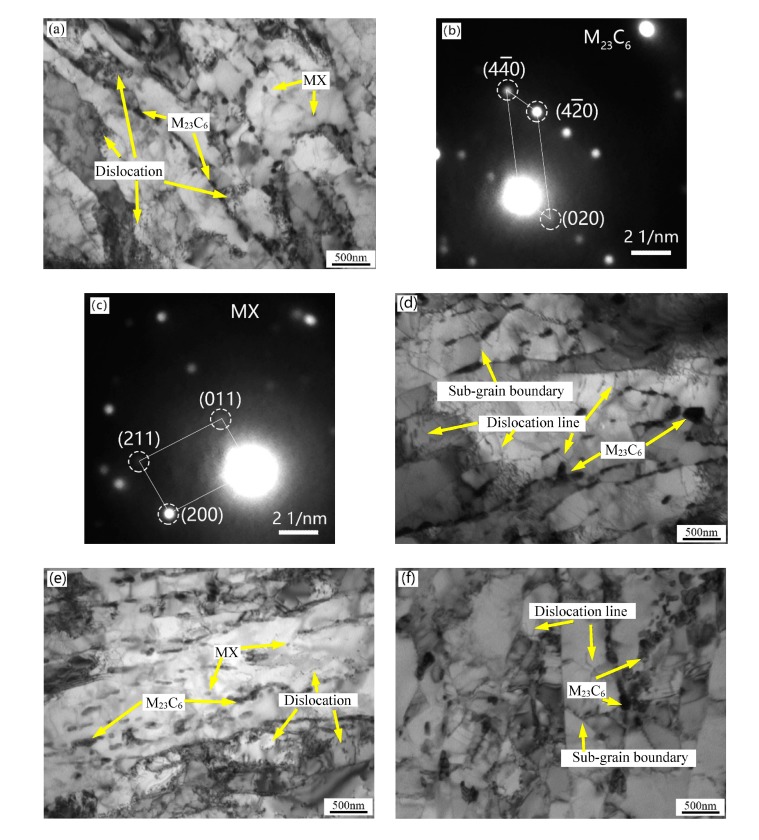
TEM images of the experimental steels after heat treatment (**a**) 1L (low Si and low temperature), (**b**,**c**) selected area electron diffraction (SAED) pattern of M_23_C_6_ and MX in 1L, (**d**) 1H (low Si and high temperature), (**e**) 2L (high Si and low temperature), and (**f**) 2H (high Si and high temperature).

**Figure 7 materials-13-00972-f007:**
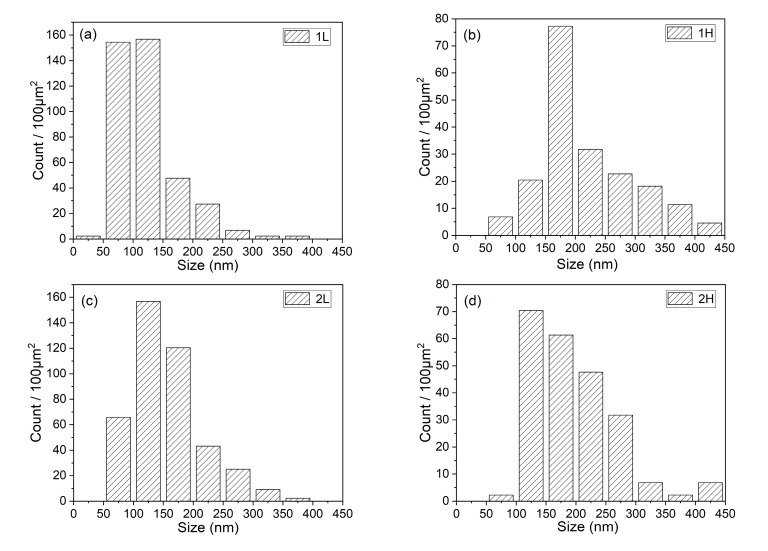
The number density of M_23_C_6_ in experimental steels after heat treatment (**a**) 1L (low Si and low temperature), (**b**) 1H (low Si and high temperature), (**c**) 2L (high Si and low temperature), and (**d**) 2H (high Si and high temperature).

**Figure 8 materials-13-00972-f008:**
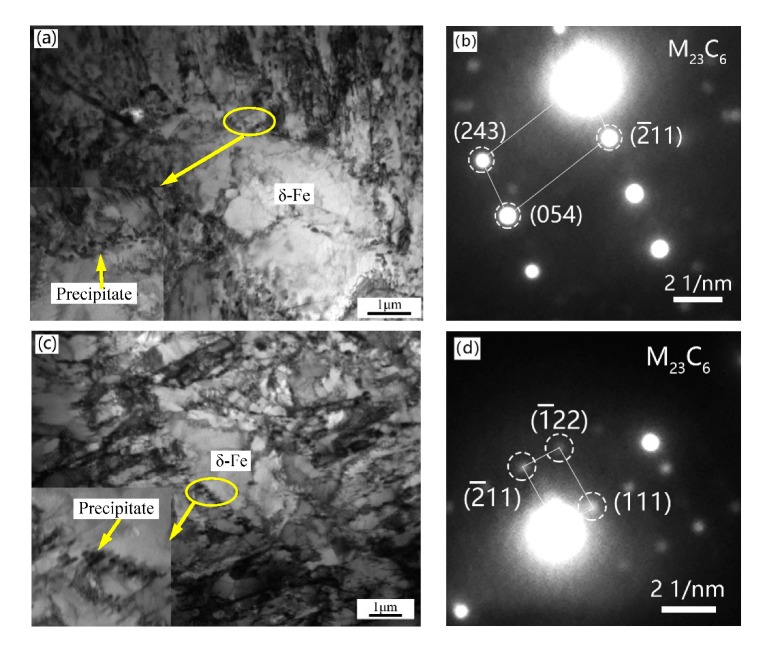
TEM images of the δ-ferrite in 2# steels after heat treatment (**a**) δ-ferrite in 2L steel (high Si and low temperature), (**b**) SAED pattern of the precipitates around the δ-ferrite in 2L, (**c**) δ-ferrite in 2H steel (high Si and high temperature), and (**d**) SAED pattern of the carbides around the δ-ferrite in 2H.

**Figure 9 materials-13-00972-f009:**
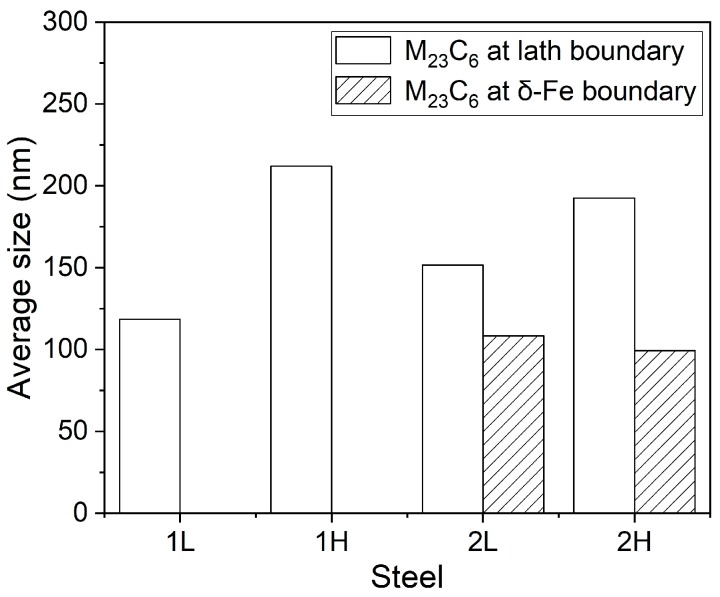
The average size of M_23_C_6_ precipitates at martensite lath boundary and δ-ferrite boundary in each sample.

**Figure 10 materials-13-00972-f010:**
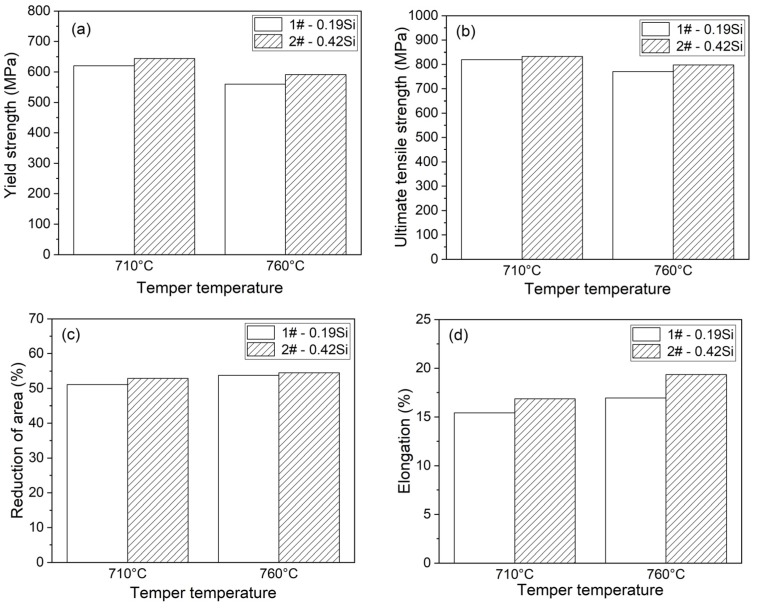
Effects of temper temperature and Si content on the mechanical properties of HT-9 steels at room temperature: (**a**) yield strength, (**b**) ultimate tensile strength, (**c**) reduction of area, (**d**) elongation, (**e**) impact energy, and (**f**) hardness.

**Figure 11 materials-13-00972-f011:**
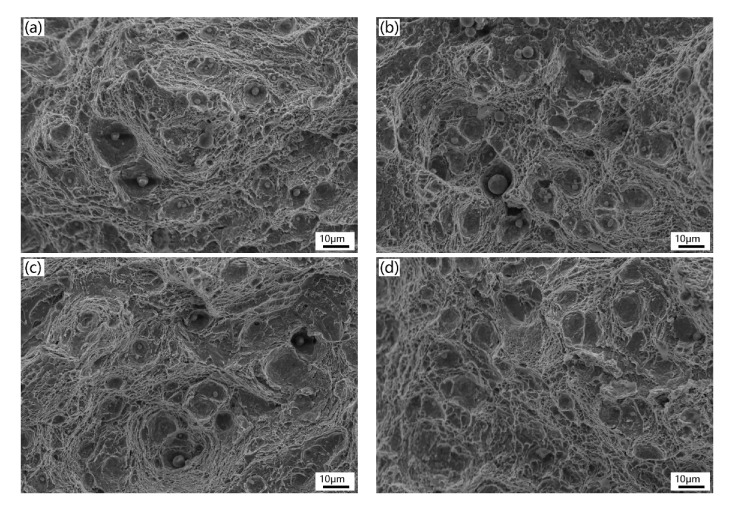
Images of impact fracture morphology: (**a**) 1L (low Si and low temperature), (**b**) 1H (low Si and high temperature), (**c**) 2L (high Si and low temperature), and (**d**) 2H (high Si and high temperature).

**Figure 12 materials-13-00972-f012:**
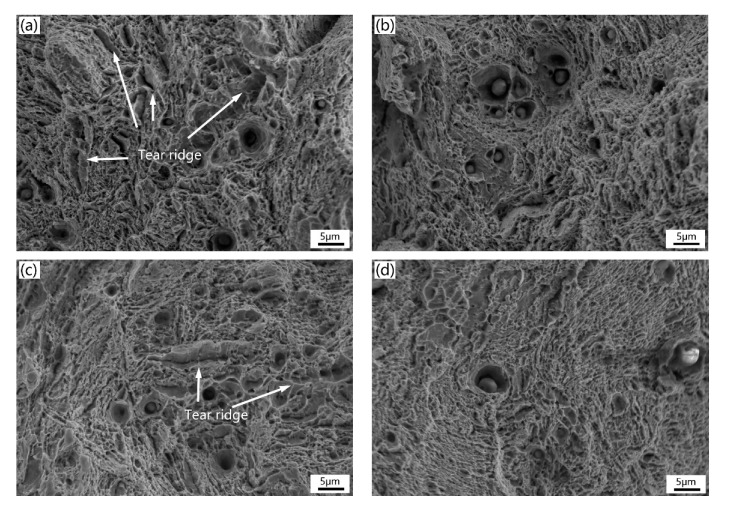
Images of tensile fracture morphology: (**a**) 1L (low Si and low temperature), (**b**) 1H (low Si and high temperature), (**c**) 2L (high Si and low temperature), and (**d**) 2H (high Si and high temperature).

**Table 1 materials-13-00972-t001:** Chemical composition of the HT-9 steel in wt.%.

Element	Fe	C	Cr	Mo	Ni	Mn	V	W	Si
1#	Balance	0.22	10.92	1.05	0.57	0.47	0.24	0.57	0.19
2#	Balance	0.21	11.62	1.06	0.55	0.47	0.26	0.6	0.42

**Table 2 materials-13-00972-t002:** Heat treatment conditions for four kinds of samples.

Samples	Type of Steel	Quenching	Tempering
1L	1#	1050 °C/60 min/air cool	710 °C/90 min/air cool
1H	1#	1050 °C/60 min/air cool	760 °C/90 min/air cool
2L	2#	1050 °C/60 min/air cool	710 °C/90 min/air cool
2H	2#	1050 °C/60 min/air cool	760 °C/90 min/air cool

## Data Availability

The raw data required to reproduce these findings cannot be shared at this time as the data also forms part of an ongoing study. The processed data required to reproduce these findings cannot be shared at this time as the data also forms part of an ongoing study.

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
