# Peer review of "Effects of Silicon Content and Tempering Temperature on the Microstructural Evolution and Mechanical Properties of HT-9 Steels"

_materials, 2020, doi:10.3390/ma13040972_

Round 1

Reviewer 1 Report

Comments to the Author:
The authors of this paper designed two kinds of HT-9 experimental steels to investigate how the Si content and the tempering temperature affect the mechanical properties of these experimental steels. It is an interesting investigation, however, some details may be considered by the authors:

Introduction

COMMENT: Page 1, line 34: I suggest the authors to add more recent references.

COMMENT: Page 2, line 44: More recent references could be added.

Results

COMMENT: Page 4, line 119: The lack of difference between different samples could be further discussed.

COMMENT: Page 6, line 119: This difference from the results of other researchers may be further commented.

The reported data are discussed and commented and the results support the authors conclusions. Therefore, I think that this paper is suitable for publication.

Reviewer 2 Report

Dear Authors,

I found your work interesting and clearly-exposed.

Best Regards.

Author Response

Dear reviewer,

I am greatly indebted to you for your review!

Best regards.

Reviewer 3 Report

[20, 27] - template, no systematics,
What is Fig. 1 for?
Figs. 1, 9, 10.
Illegible scales.
Fig. 3 - illegible graduations and markings,
in 2 # (0.42Si) steel sample - 0.42 Si ...,
no phase marks on some Figs.
Hardly legible phase markings on some Figs.
100nm - 100 nm,
Conclusion 2 is poorly represented in paper.
